# Use and Abuse of Electrocautery in Adenoidectomy Hemostasis

**DOI:** 10.3390/medicina59040739

**Published:** 2023-04-10

**Authors:** Veronica Epure, Razvan Hainarosie, Catalina Voiosu, Dan Cristian Gheorghe

**Affiliations:** 1ENT Department, Carol Davila University of Medicine and Pharmacy, 020021 Bucharest, Romania; 2ENT Department, “MS Curie” Hospital, 077120 Bucharest, Romania; 3I.F.A.C.F.–ORL Prof. Dr. D. Hociota, 061344 Bucharest, Romania

**Keywords:** bipolar electrocautery, adenoidectomy, postoperative pain, neck pain, malodor, velopharyngeal insufficiency

## Abstract

*Background and objectives:* Bipolar electrocautery is commonly used to control bleeding after cold-instrument pediatric adenoidectomy, but the surgeon should be aware of the possible side effects. OBJECTIVE: The aim of our study is to investigate the effects of bipolar electrocautery when used for bleeding control at the end of an adenoidectomy procedure. *Materials and Methods*: We evaluated the effect of electrocautery on postoperative pain, velopharyngeal insufficiency symptoms, postoperative nasal obstruction, and rhinorrhea in a group of 90 children undergoing adenoidectomy in our ENT department over a period of 3 months. *Results*: After statistically analyzing the data, we found that the duration of postoperative pain, the duration of rhinorrhea and nasal obstruction, and the duration of painkiller administration, as well as the velopharyngeal insufficiency symptoms, were significantly longer in patients in whom electrocautery was used for hemostasis. A significantly higher incidence of posterior neck pain and halitosis (oral malodor) was noted in the patients in whom electrocautery was used for adenoidectomy hemostasis. *Conclusions*: Bipolar electrocautery use should be limited during pediatric adenoidectomy hemostasis because of the possible side effects: longer postoperative pain, prolonged nasal obstruction, rhinorrhea and velopharyngeal insufficiency, and halitosis. We noted some side effects that were specific to electrocautery use during adenoidectomy: posterior neck pain and oral malodor. Acknowledging the risk for these symptoms can help to alleviate the anxiety of both the parents and the patients regarding the expected postoperative outcomes.

## 1. Introduction

Adenoidectomy and tonsillectomy continue to be the most commonly performed procedures in pediatric ENT [1]. Despite advances in surgical techniques, post-adenoidectomy hemorrhage remains the most important cause of morbidity for children undergoing adenoidectomy, especially in the early postoperative period. A neat hemostasis at the end of the procedure is important. Tonsillectomy and adenoidectomy are frequently indicated for obstructive sleep apnea or recurrent infections of the tonsils or adenoid mass [1,2,3]; the incidence of adenotonsillectomy has risen dramatically over the years due to its efficacy in preventing obstructive sleep apnea and its negative side effects (cognitive and behavioral disorders, nocturnal enuresis, learning difficulties, and hypertension) [3,4].

The endeavor to decrease postoperative pain and minimize the risk of postoperative hemorrhage has brought attention to novel adenoidectomy techniques and instrumentation. To date, different methods have been used to perform adenoidectomy: cold surgery, using the conventional curettage with a Beckmann curette or microdebrider or endoscopic-assisted adenoidectomy, and “hot methods”, with either bipolar electrocautery or radiofrequency (coblation) [5,6]. In 2016, electrocautery adenoidectomy was considered the most common adenoidectomy technique used in the United States [6].

Hemostasis after cold-instrument adenoidectomy is usually spontaneous, aided by the topical use of epinephrine solutions, antifibrinolytic agents (tranexamic acid), vasoconstrictors, hydrogen peroxide, saline solutions at different temperatures, or cold air flow by continuous aspiration [1,7,8,9]. In the case of hemostasis failure, posterior nasal packing or electrocoagulation (both performed under general anesthesia at the end of the procedure) can be used to achieve it. Locally applied bipolar electrocautery is preferred to posterior nasal packing [1] to avoid significant morbidity during the postoperative period.

Electrosurgical units are a common form of electrical equipment in the operating theatre.

In avoiding unnecessary damage to surrounding tissue, bipolar electrosurgery use seems better due to a more limited area of thermal spread compared with that of monopolar cautery [8,9,10,11,12]. The explanation lies in the concentrated current density applied through bipolar forceps with a limited coagulation of the small amount of tissue contained between the tips of the forceps and a minimal effect on the surrounding tissue [10]. Because the path of least resistance is the shortest distance between the two electrodes, the probability of the current travelling via an alternate pathway is reduced in the case of bipolar cautery.

Monopolar cautery is rarely used during an adenoidectomy for adenoid mass removal and/or hemostasis [13,14,15]. Most authors agree that bipolar cautery is safer to use and that the tissue damage is more predictable compared to that of monopolar cautery. Studies report that tissue necrosis and the extent of thermal damage to the tissues are greater in the case of monopolar cautery compared to that of bipolar cautery or radiofrequency [16,17]. When using bipolar electrocautery, the maximal thermal spread should be within 5mm of the tissue depth and limited to the mucosal layer.

The aim of our study is to investigate the effects of bipolar electrocautery and its specific side effects when used to control bleeding at the end of adenoidectomy procedures. The adenoidectomy procedure is performed in our department with cold instruments, with bipolar electrocautery applied locally only when spontaneous hemostasis fails. Written informed consent from all the parents of the children enrolled in this study, both for surgical treatment and for inclusion in this study, was obtained.

## 2. Materials and Method

We conducted an observational prospective study (following the STROBE Statement checklist for reporting observational studies) on 90 consecutive pediatric patients undergoing adenoidectomy over a period of 3 months in our ENT department (November 2022–January 2023). From the number of children undergoing adenoidectomy in our operating theatre, two groups of patients were randomly selected: group A (the group with electrocautery; 45 patients, in whom we used bipolar electrocauterization in order to control post-adenoidectomy bleeding) and group B (the group without electrocautery; 45 patients with spontaneous post-adenoidectomy hemostasis).

Our hospital’s ethics committee approved this clinical study (no. 53792/12 December 2022).

Initially, 100 randomly chosen children aged 0 to 18 years were enrolled in our study if they met the following inclusion criteria: 1—admission to the pediatric ENT department of M.S. Curie Children’s Hospital between November 2022 and January 2023 (on 2 randomly chosen days per week); 2—indication of adenoidectomy under general anesthesia with oral intubation (due to obstructive sleep apnea, recurrent upper respiratory tract infections, recurrent acute otitis media, and chronic serous otitis media); 3—laboratory tests (blood markers), performed preoperatively by our hospital’s laboratory, showing no major coagulation abnormalities.

All the surgical procedures that took place in the operating theatre were recorded in the surgical records saved in the hospital’s database. All the adenoidectomies were performed by 3 surgeons (consultants with over 20 years of pediatric ENT practice), using the same technique and the same type of instruments. Classical cold instruments were used in all the adenoidectomies (Beckmann’s steel curette with dimensions corresponding to the child’s nasopharynx). Adenoidectomy was performed with direct vision of the nasopharynx (hyperextension of the child’s neck and the unilateral Nelaton’s probe were used to retract the soft palate cranially for enhanced visualization). Digital palpation of the adenoid mass and nasopharynx was routinely performed prior to and after the procedure in order to ensure complete excision of the adenoids. After complete resection of the adenoid mass, hemostasis followed the same steps: a cotton packing was introduced into the nasopharynx twice and left in place until soaked in blood, followed by one packing soaked with a vasoconstrictor agent (epinephrine dilution 1/10). Sometimes, direct irrigation with saline solution at room temperature, hydrogen peroxide applied locally, or cold air suctioning were used. In some patients with failure of hemostasis (considered as such in cases with over 10 min hemostatic time or the failure of the packing cottons to stop bleeding), bipolar electrocautery was used to complete the hemostasis; these patients were enrolled in group A. In our department, we use bipolar electrocautery for hemostasis after adenoidectomy; so, the present study refers to its use during the procedure. After surgery, the children went to the postoperative intensive care unit for 20 min monitoring; then, they returned to the clinical ward when stable, when there was no bleeding, and were able to start the oral intake of clear liquids. Early oral intake was encouraged.

The parents of the patients in both group A and group B were given questionnaires (Appendix A) at the discharge from hospital, which they returned completed at the routine postoperative follow-up appointment after 3 weeks. These contained the following questions: how many days the postoperative pain (i.e., odynophagia/retronasal pain) last? How many days did the parents administer painkillers to the child? How many days did nasal obstruction and/or rhinorrhea last? Was there any ear pain (Yes/No answer)? Was there any neck pain (posterior headache) (Yes/No answer)? Was there any foul smell (malodor) or undesirable taste present (Yes/No answer)? How many days did the velopharyngeal insufficiency symptoms last (hypernasal speech)? Initially, the questionnaires were given to 100 parents of adenoidectomy children at the discharge from hospital (printed on blue paper for group A children and on white paper for group B children), but 10 were lost to the follow-up (they did not show up at the 3-week postoperative control, or they did not return the completed questionnaires).

For pain assessment in small children (under 4 years of age), the parents were advised to use a visual analogue scale with happy/sad faces, drawn on the back of the questionnaires, when necessary, in addition to their behavioral assessment of the child’s wellbeing during their first few postoperative days. The responses were registered as a Yes/No answer; the precise grading of pain intensity was not taken into account in order to eliminate the bias resulting from the subjectivity of the children’s caretakers.

It is important to mention that in the first question of the questionnaire, postoperative pain refers to odynophagia and/or retronasal pain, while the question “was there any neck pain/posterior headache?” specifically refers to the pain the patients feel at the back of their head; this is the pain which is referred from the adenoid area to the posterior neck region. Ear pain after adenoidectomy is also referral pain.

All the variables from the questionnaires were entered into an Excel database (age and sex of each patient; duration of postoperative pain in days; duration of postoperative painkiller medication; duration of nasal obstruction and/or rhinorrhea in days; duration of velopharyngeal insufficiency symptoms; presence or absence of ear pain; and posterior headache and fetidity). In this study, the size of the adenoid mass was not graded or taken into account. The patient records were anonymized and analyzed. Statistical analysis was performed using the R project for statistical computing (R version 4.1.3); the Welch two-sample *t*-test and the Pearson chi-squared test with Yates continuity correction were used to compare the results between the two groups; a value of *p* ˂ 0.05 was considered statistically significant.

## 3. Results

The Welch two-sample t-test was used to compare the variables between the two groups—group A (45 patients, in whom we used bipolar electrocauterization in order to control post-adenoidectomy bleeding) (electrocautery group) and group B (45 patients with spontaneous post-adenoidectomy hemostasis) (no electrocautery group). In all the patients, adenoidectomy was performed with cold steel instruments; bipolar electrocautery, if any, was used during hemostasis only in the children from group A.

In the analysis of the patients’ age, the mean age and distributions were similar between the two groups—the mean age in group A was 4.69 years (1–16 years), and in group B, it was 4.73 years (1–12 years); the difference was not statistically significant (*p* = 0.925); thus, the groups were similar and comparable. Group A consisted of 24 girls and 21 boys, while group B consisted of 15 girls and 30 boys (33% girls and 66% boys).

In comparing the values for the duration of the postoperative pain, there was a statistically significant difference between the groups. The mean duration of postoperative pain in group A (electrocautery group) was 4.62 days (1–14 days), while in group B it was 2.13 days (with variations between 1 and 7 days). The difference was statistically significant (*p* < 0.001) (Figure 1).

In comparing the values for the duration of painkiller administration, there was a statistically significant difference between the groups. We usually recommend 3 days of analgesic medication after simple adenoidectomy in children. The mean duration of postoperative medication was 3.96 days for the patients in group A (electrocautery group) and 3.16 days for the patients in group B (no electrocautery). The difference was statistically significant (*p* = 0.044) (Figure 1).

In comparing the values for the duration of rhinorrhea and/or postoperative nasal obstruction in the adenoidectomy patients, there was a statistically significant difference between the groups. The mean duration of rhinorrhea/nasal obstruction was 3.09 days for the patients in group A (with the use of electrocautery) and 1.98 days for those in group B (without electrocautery). The difference was statistically significant (*p* = 0.015) (Figure 2).

In comparing the values for the duration of post-adenoidectomy velopharyngeal insufficiency symptoms (hypernasal speech, in days), there was a statistically significant difference between the groups. The mean duration of velopharyngeal insufficiency was 3.71 days for the patients in group A (electrocautery) and 1.64 days for the patients in group B (no electrocautery). The difference was statistically significant (*p* < 0.001) (Figure 2).

In comparing the frequency of postoperative malodor (halitosis) encountered in the patients in group A (31 out of 45 patients) to that of the patients in group B (1 out of 45 patients), there was a statistically significant difference between the groups (X-squared = 40.781, df = 1, *p* < 0.001. Postoperative halitosis was encountered more frequently in the patients in whom electrocautery was used intraoperatively (Figure 3).

In comparing the frequency of postoperative ear pain encountered in the patients from group A (6 out of 45 patients) to that of the patients in group B (4 out of 45 patients), the difference between the groups was not statistically significant (X-squared = 0.112, df = 1, *p* = 0.737) (Figure 4).

In comparing the frequency of postoperative posterior headache (neck pain) encountered in the patients from group A (34 out of 45 patients) to that of the patients in group B (2 out of 45 patients), there was a statistically significant difference between the groups (X-squared = 44.491, df = 1, *p* < 0.001). Postoperative neck pain (pain at the back of the child’s head) was more frequently encountered in the patients in whom electrocautery was used intraoperatively (Figure 5).

No Griesel’s syndrome was noted in our group A patients; there was only a case of intense posterior headache and neck stiffness on the 2nd day post-adenoidectomy in a 6-year-old male child; this subsided in 48 h under intravenous antibiotics and cortisone.

The results and structure of our studied groups are summarized in Table 1.

## 4. Discussions

There is conflicting information in the literature regarding electrosurgery wound healing in oral or ENT surgery [15,18]. Most of the authors report that electrocautery damage spreads to large areas and creates burning wounds as well as delayed epithelization of the surgical bed. Regarding the delayed side effects of cauterization, most authors agree that it induces scar tissue formation with a risk of nasopharyngeal stenosis [19,20].Some authors mention the presence of undesirable taste and smell in the patient in the immediate postoperative period and retropharyngeal edema, prolonged velopharyngeal insufficiency (hypernasal speech), increased neck pain (12%), and even Griesel’s syndrome (nontraumatic atlantoaxial subluxation after an inflammatory process in the upper cervical region) [20,21] following electrocautery use in adenoidectomy. Others state that a modified technique of bipolar tonsillectomy or adenoidectomy can lead to reduced postoperative pain [22,23]. 

In our study, we found significantly increased postoperative pain, prolonged velopharyngeal insufficiency, prolonged rhinorrhea, and nasal obstruction and the increased need for analgesic medication in the immediate postoperative period in children in whom bipolar cautery was used for hemostasis. Our findings are consistent with the other data from the literature [22,23,24,25,26].

The aim of our study was to draw the attention of the ENT surgeons to these negative effects following cautery use in adenoidectomy, based on the perceptions of the patients’ caretakers of the possible postoperative course.

As with the findings in other literature reports [22,23,24,25,26], bipolar cautery use for hemostasis in adenoidectomy patients in our study increased the duration of postoperative pain. Although on an individual basis these small changes in the duration of postoperative pain (from a mean of 2.13 days to 4.62 days) may seem to have little clinical significance, the difference was statistically significant (*p* < 0.05). At the same time, the postoperative antialgic medication need was also significantly prolonged by cautery use in adenoidectomy (3.96 days for the patients in the electrocautery group compared to 3.16 days for the patients in the group without it).

A possible hypothesis to explain the exacerbation of postoperative pain after electrocautery use in adenoidectomy can be found in the literature. Most authors agree that electrosurgical incisions appear to produce more inflammatory responses and tissue destruction compared to cold-cut wounds [11,12,17]. This increased local inflammation could also explain the prolonged duration of rhinorrhea and nasal obstruction in the group A patients enrolled in our study (for a mean of 1.98 days to 3.09 days).

Temporary velopharyngeal insufficiency is a well-recognized, yet rare, complication of adenoidectomy [26,27,28]. Its symptoms may include hypernasal speech and food regurgitation through the nasal passages during feeding. We found literature reports of a direct association between electrocautery use and postoperative velopharyngeal insufficiency [26]. In our study, the mean duration of velopharyngeal insufficiency symptoms after adenoidectomy was significantly increased in bipolar cautery patients—3.71 days for patients in group A (electrocautery) compared to 1.64 days in group B (no electrocautery).

Halitosis (oral malodor) is another possible complication of adenoidectomy, especially when using electrocautery [27,28,29]. Oral malodor in children has been attributed to several oral and extra-oral etiologies. It has been associated with the presence of an infected adenoid mass; gastrointestinal, dental, or respiratory conditions; and systemic conditions [27,28,29]. One study reports the presence of halitosis for up to 7 days post-adenoidectomy (with bipolar cautery use), with a maximum intensity on days 2 and 3 after surgery. The same study reports the persistence of postoperative pain after electrocautery adenoidectomy for up to 7 days postoperatively (with a peak on day 3), independently of the adenoid mass hypertrophy grade [28].

Our study noted that postoperative malodor is more frequently found (68.89%) in patients in whom bipolar cautery has been used and that it can persist postoperatively for up to 10 days. Our results are consistent with the other published data [27,28,29].

According to some authors, all electrosurgical lesions demonstrate some areas of desiccation, coagulation, and carbonization, regardless of the power or the waveform used [18,19]. The nasopharyngeal inflammatory reaction and tissue necrosis could account for the postoperative halitosis. Electrosurgical currents have two possible effects on tissues: cutting or coagulation. The coagulation effect is used for bleeding control during adenoidectomy hemostasis. Coagulation implies the raising of the temperature of the living tissues from approximately 37 °C to above 45 °C in order to achieve the coagulation of the protein content of the cells; it is an irreversible state. When tissue temperature rises to over 60 °C, the water content of the cell is driven out and desiccation begins (up to 100 °C); the desiccation type of coagulation is the therapeutic goal for accomplishing hemostasis [10,11,12,17]. Beyond desiccation, the continued application of heat causes the disintegration of cellular components into oxygen, nitrogen, hydrogen, and carbon (carbonization or black coagulation).

Ear pain (earache) is sometimes mentioned by different authors as a possible consequence encountered after adenoidectomy (or tonsillectomy) [23,24]. As most authors do not link the presence of ear pain after surgery to bipolar cautery use, we tried to analyze the presence of postoperative ear pain in our groups of patients. However, the two groups showed an insignificant difference in postoperative ear pain (*p* > 0.05). We can conclude that ear pain is not specific for electrocautery use during the first days following adenoidectomy.

The presence of a posterior headache (referral pain from the site of the surgery) is frequently mentioned by some authors after electrocautery adenoidectomy [21,22,25,28]. Comparing the frequency of postoperative neck pain encountered in patients from group A (34 out of 45 patients, i.e., 75, 55%) to that from group B (2 out of 45 patients), we found a statistically significant difference between the groups (*p* < 0.05); we can conclude that postoperative neck pain is more frequently found in patients in whom electrocautery was used for hemostasis. These findings are consistent with those of other literature reports [21,22,25,28].

According to our results, neck pain and the presence of halitosis are quite specific for electrocautery used to control bleeding at the end of adenoidectomy (statistically significant) (occurring in 69% and 75% of these patients, respectively). Griesel’s syndrome (atlanto-axial subluxation due to posterior neck inflammation and rigidity) is a rare but possible occurrence in adenoidectomy children [20,21], but there were no Griesel’s syndrome cases in our enrolled patients.

Knowing that symptoms of pain, malodor, and nasal obstruction are expected to persist longer in patients in whom cautery is used, we can recommend analgesics and nasal vasoconstricting agents for up to 5 or 7 days (instead of the usual 3 days), and we can advise the parents of these children that oral malodor and posterior neck pain are possible occurrences.

Many studies compare the results of cold (curettage) adenoidectomy to electrocautery or radiofrequency adenoidectomy (this means the ablation of the adenoid mass using bipolar electrocautery or radiofrequency coagulation—aspiration). Most of the authors agree that the operative time and blood loss are significantly reduced in the case of cautery use [5,6]. There is confusing information regarding the postoperative complications of electrosurgical adenoidectomy. Whereas some authors report no complications after bipolar cautery adenoidectomy [23,25], others report a higher risk of secondary (delayed) hemorrhage and a greater incidence of neck pain after electrocautery use (incidence up to 12%) [6], while coblation and radiofrequency (using lower temperatures for tissue removal and coagulation) seem to produce less damage to underlying tissues, with lower rates of neck pain. Other authors have reported long-term complications after electrosurgical adenoidectomy, such as nasopharyngeal burns, scarring, and stenosis [23,25].

Adenoid regrowth was reported by some authors to have a lower incidence after electrocautery adenoidectomy (2.8%) compared to the reported 5.4% after cold adenoidectomy [28]. Our study did not take into account these long-term effects of electrocautery on surgical results.

A study of 15,734 patients found the incidence of secondary hemorrhage after adenotonsillectomy to be 2.8 times higher after cold dissection plus bipolar hemostasis, 3.2 times higher after coblation, and 4.3 times higher after diathermy scissors, compared to cold technique [6]. No primary or secondary hemorrhage occurred in our patients, in either the electrocautery group or the no cautery group. This may be due to the small number of patients in our cohort.

Most authors state that the degree of tissue damage from electrocautery depends on the duration of the energy applied, the power level used, and the tissue properties (vascular density and fat content) [18,19]. These mentioned studies addressed electrocautery adenoidectomy techniques with the prolonged use of cautery to achieve adenoid mass ablation; our study researched the side effects related to the limited use of electrocautery during adenoidectomy hemostasis (intermittent use of bipolar electrocautery at 28–35 W). We found negative effects even after limited use of bipolar cautery for bleeding control after adenoidectomy.

Studies performed on fresh cadavers using a thermistor to evaluate temperature changes in the prevertebral fascia after electrocautery use during adenoidectomy (30 Watts for a period of 30 s—a relatively short period of time) show peak readings averaging up to 74 °C (with mean temperatures of 51.8 °C) [21]. We can presume that electrocautery use during adenoidectomy, regardless of the duration of usage, produces notable side effects. This is consistent with the findings of other authors [19] and could account for the prolonged pain, malodor, and velopharyngeal insufficiency in the patients in whom cautery was used intraoperatively.

The present study might have some limitations. The fact that the study was based on a single-center experience and that only the side effects occurring in the first 2 weeks were taken into account (no long-term effects of electrocautery use such as nasopharyngeal scarring or adenoid regrowth rate) may have skewed the results in unpredictable ways. The dimension of the sample of patients enrolled may seem limited, but there were enough data to be statistically analyzed and to generate possible conclusions. The difficult communication with small children (our groups’ mean ages were 4.69 and 4.73 years, respectively) could account for the inaccuracies in assessing the question “For how long did the pain last postoperatively?”, even though a visual analogue scale was used for smaller children [30,31,32,33,34]. It is also possible that some parents did not follow the instructions and did not give sufficient analgesic medication to their children in the first few postoperative days, resulting in reports of higher pain levels. The effects of electrocautery use may need further investigation with larger groups of patients.

## 5. Conclusions

Bipolar electrocautery should be used with caution during adenoidectomy hemostasis. Electrocauterization stops bleeding during adenoidectomy but produces swelling, oedema, and significant pain after surgery, along with prolonged postoperative pain, prolonged rhinorrhea and nasal obstruction, prolonged velopharyngeal insufficiency symptoms, and a prolonged need for analgesic medication. The presence of oral malodor and neck pain were noted in the children in whom electrocautery was used for hemostasis during adenoidectomy. Understanding these symptoms can help in providing advice for parents of adenoidectomy children on the possible expected outcomes and to reduce the anxiety of both the child and his/her parents during the postoperative period.

## Figures and Tables

**Figure 1 medicina-59-00739-f001:**
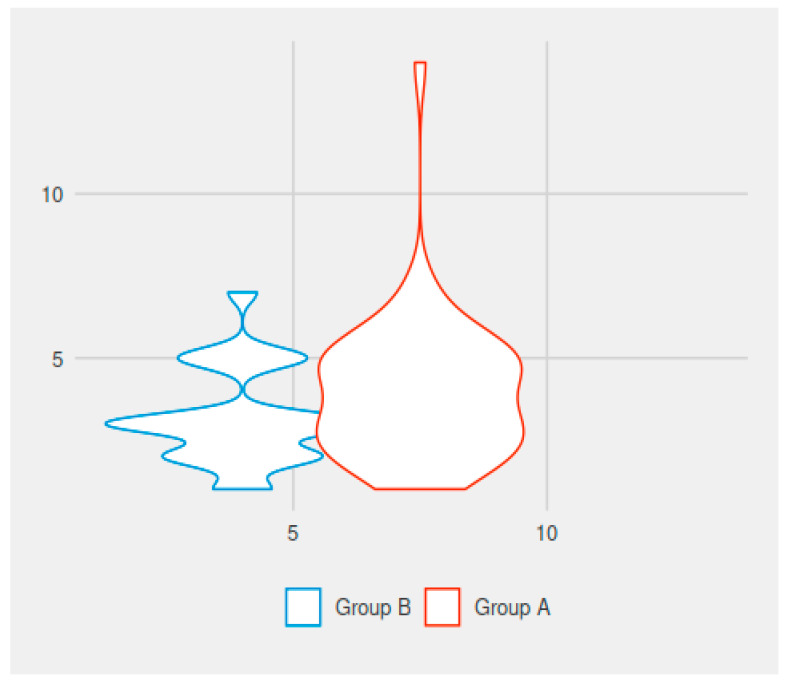
Duration of postoperative pain versus duration of antialgic administration with or without electrocautery use (on the vertical line—the duration of postoperative pain in the two groups of children, in days; on the horizontal line—the duration of painkiller medication need in the two groups of enrolled children, in days).

**Figure 2 medicina-59-00739-f002:**
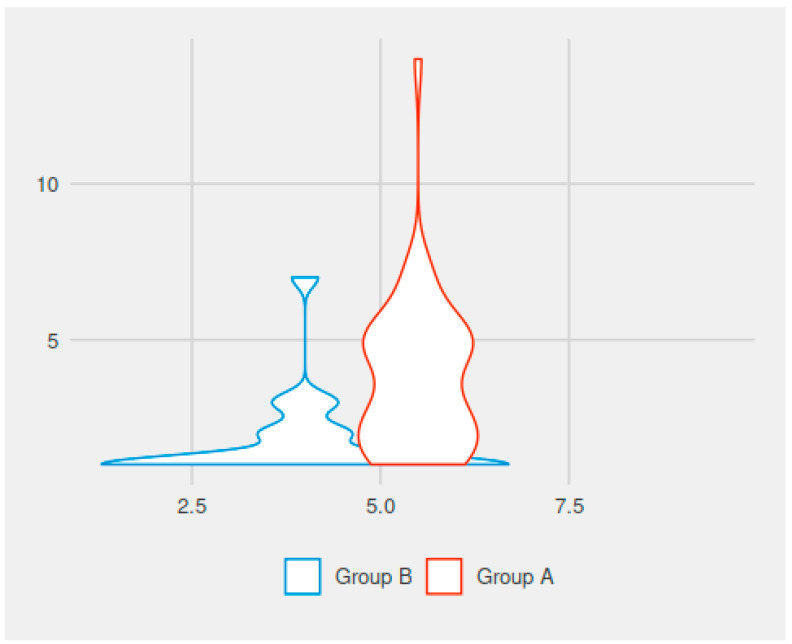
Duration of rhinorrhea and/or nasal obstruction versus velopharyngeal insufficiency with or without electrocautery use (on the vertical line—the duration of rhinorrhea and/or postoperative nasal obstruction in the two groups of children, in days; on the horizontal line—the duration of velopharyngeal insufficiency symptoms in the two groups of enrolled children, in days).

**Figure 3 medicina-59-00739-f003:**
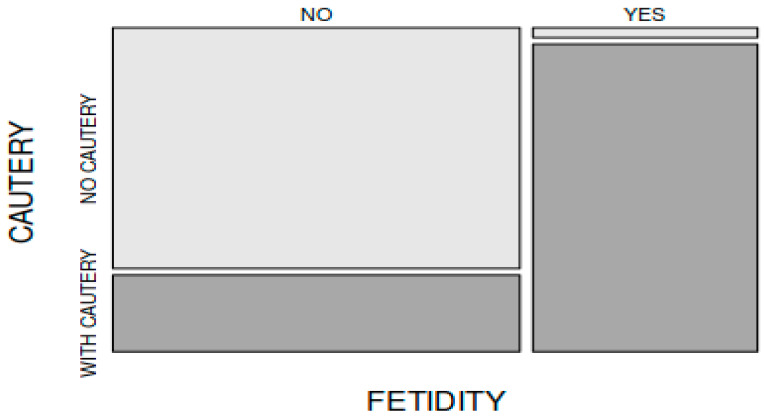
Incidence of postoperative fetidity occurrence (halitosis) was significantly higher in group A patients (electrocautery group).

**Figure 4 medicina-59-00739-f004:**
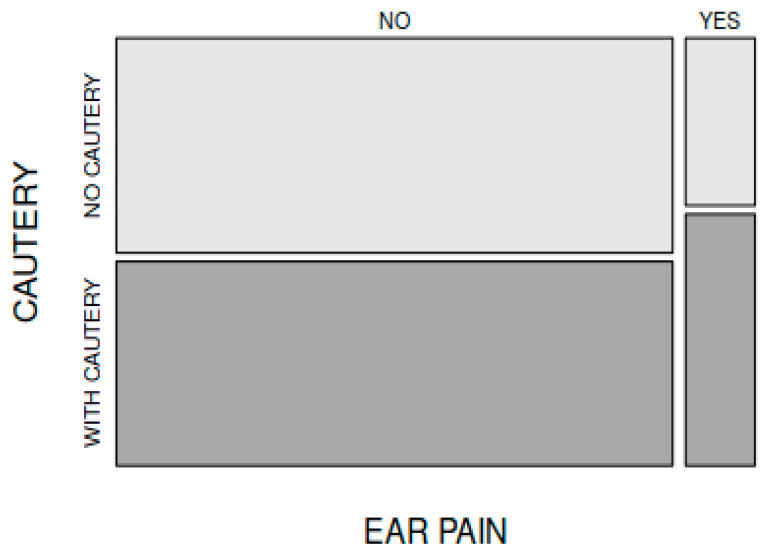
Incidence of postoperative ear pain was similar regardless of electrocautery use.

**Figure 5 medicina-59-00739-f005:**
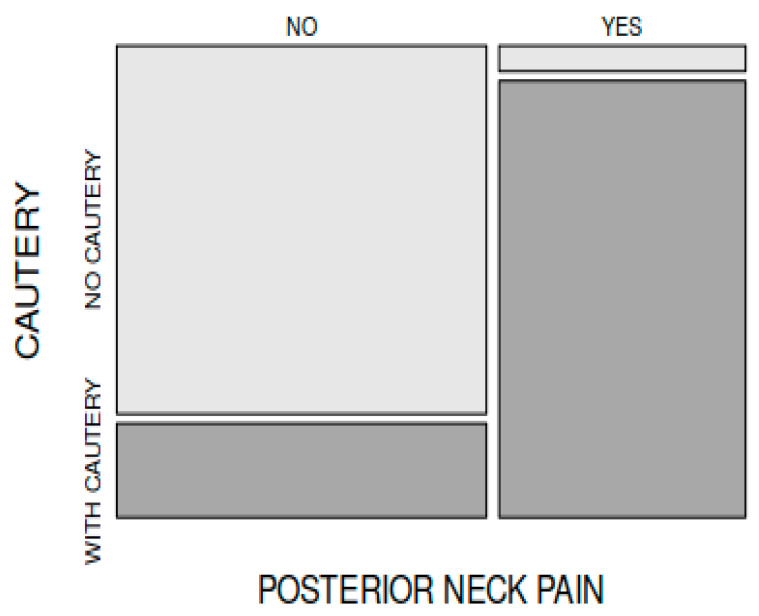
Incidence of posterior neck pain was significantly higher in patients in group A (electrocautery group).

**Table 1 medicina-59-00739-t001:** Structure of two groups, variables studied, and their statistical significance: ss—statistically significant (*p* < 0.05), ns—not significant.

	Group A	Group B	*p* Value
Mean age of patients	4.69 years (1–16)	4.73 years (1–12)	*p* = 0.925 (ns)
Sex of patients	24 girls	15 girls	
	21 boys	30 boys	
Duration of postoperative pain	4.62 days (1–14)	2.13 days (1–7)	*p* < 0.001 (ss)
Duration of postoperative antialgic administration	3.95	3.16	*p* = 0.044 (ss)
Duration of postoperative nasal obstruction/rhinorrhea	3.09	1.98	*p* = 0.015 (ss)
Duration of velopharyngeal insufficiency	3.71	1,64	*p* < 0.00 (ss)
Presence of halitosis	31/45	1/45	*p* < 0.001 (ss)
Presence of postoperative ear pain	6/45	4/45	*p* = 0.737 (ns)
Presence of postoperative posterior headache	34/45	2/45	*p* < 0.001 (ss)

## Data Availability

Data available on request to the corresponding author.

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
