# Peer review of "Use and Abuse of Electrocautery in Adenoidectomy Hemostasis"

_medicina, 2023, doi:10.3390/medicina59040739_

Round 1
Reviewer 1 Report
An interesting study regarding bipolar electrocautery in cold instruments pediatric adenoidectomy.
The study was conducted on odynophagia/retro nasal pain, pain-killers administration, nasal obstruction and/ or rhinorrhea, ear pain/neck pain (posterior headache), foul smell or undesirable taste, velopharyngeal insufficiency (hypernasality speech). The last postoperative appointment was at 3 weeks. Eighter there were no (assuming) postoperative bleeding it should be specified in the text (regardless there were not in the questionnaire).
Author Response
There was no early or delayed postoperative bleeding in our studied patients, due to small sample size (a total of 90 children) – it is specified in DISCUSSION section, in paragraph 16. Larger samples of patients are needed in order to assess the incidence of bleeding after electrocautery use in adenoidectomy children.

Reviewer 2 Report
Dear authors, I have carefully read your article which draws attention to an important aspect that has not been sufficiently evaluated in the literature.
- 45 patients per group were selected. Is the same number between the two groups a coincidence? We are talking about 100 patients, 10 lost to follow up, with an exact 50% split between groups. Did you consider all adenoidectomies performed in the observation period or or did you select the patients to make the numbers between the two groups comparable?
- you could include in the discussion that the patients in the electrocautery group are patients who have bled more, in which other hemostatic treatments have not been sufficient. This could be a selection bias.
- Below which what age was the visual analogue scale used for pain? In how many patients?
- There are some paragraphs that do not correspond to the chapters: lines 107-111 should belong to materials and methods, lines 271-312 in the discussion are actually results.
- I would express the statistical significance with 3 decimal digits (for example p=0.044 or p<0.001), both in the text and in the table.
- I don't find figures 1 and 2 very clear, you could specify the variable on the axes.
Author Response
The initial 100 enrolled patients were randomly selected (from a larger number of patients undergoing adenoidectomy in our department over a 3-months period), a few of each group were lost to follow-up and excluded from the study, but we chose equal number of patients in each group on purpose, in order for the groups to be similar and comparable; we decided 45-50 patients in each group is enough for statistical analysis and generalisable conclusions, but of course the study can be extended. Practically, before hospital discharge, to some randomly chosen adenoidectomy patients (a total of 100) were given blue or white questionnaires (blue to those in whom electrocautery was used, white in no-cautery patients; containing the same questions and happy / sad faces for pain assessment) and they were asked to sign the consent for inclusion in the study and to return the questionnaires completed at their 3-weeks follow-up control.
We advised patients of small children to use the visual analogue scale (drawn on their questionnaires) in children under 4 years of age, to complete their behavioral assessment of the child’s postoperative distress level if needed, but in the end the exact scores (0-2-4-8-10) for pain level were not taken into account, we registered only the number of days of postoperative pain for each child and compared the means between the two groups.
We have excluded lines 271-312 as such from the discussion section and have added some explanations and references explaining our results.
We have expressed the statistical significance with 3 decimal digits as suggested.
We have specified the variables on the axes for figures 1 and 2 (see figure legend below each figure).

Reviewer 3 Report
The subject of the study is interesting.
Nevertheless the way of presentation and the language need improvement. In many cases, the sentences are too long that make them difficult to understand. Fetidity – I'd rather use the term "halitosis" or “oral malodour.”
Intoduction is chaotic. I would advise to rearrange this section: first - the medical problem and the surgical procedure, then complications etc.
Lines 54-55 : radiofrequency and electrocautery? Do authors regard them the same method? It should be clearly stated in this section, not only explained in the discussion.
Lines 82-86: Where is monopolar electrocautery used in these procedures?please add the necessary references.
Lines 106-109: long sentence containing a few plots. A clear language is needed.
Methods:
Lines 135 and 137 : repetition of the term “procedures”
Lines 137-143: long sentence, difficult to follow the idea, grammar.
Line 154: grammar…
Questionnare should be attached to the manuscript , not only described.
The arrangement of the pain analogue scale is not clear.
How was the velopharyngeal insufficiency assessed? Were parents' opinions reliable regarding to hypernasal speach assessment?
Results:
Lines 194-198: mixed tensed used. In my opinion past tens is preferred during the presentation of the results.
Why was the visual analogue scale mentioned in methods section if not mentioned in results?
Discussion:
Common language (eg on the long run..)
Line 270: the term data is a plural verb….
Lines 266-306: It is just the repetition of results.
The number of references used in discussion is extremely low.
The manuscript is unacceptable for publishing in Medicina in this form.
Author Response
The article has gone through English proofreading and was reviewed by a native speaker, we have corrected the grammar inaccuracies following the suggestions.
The term “fetidity” has been replaced by halitosis or oral malodour.
The introduction has been shortened and some references have been added. Line 54-55: radiofrequency and electrocautery are different methods, now it is clearer stated in the text.
In lines 82-86 some explanations about monopolar cautery use in adenoidectomy have been added.
Grammar problems and inaccuracies in lines 106-109, 135, 137, 137-143, 154, 194-198, 270 have been corrected.
The questionnaire form has been added to the manuscript. The visual analogue scale was only an aid for the parents to complete behavioral assessment of postoperative pain in their children (we thought it would be helpful in smaller children), the exact values of pain level was not taken into account, only the presence or absence of pain and the duration of it (in days).
The velopharyngeal insufficiency was explained by the ENT specialist to the parents before hospital discharge (as hynernasality speech), showing examples of children with hypernasal speech (either their own children or videos with this issue) and assessed by each parent in the first few days postoperatively; the parents’opinions seemed reliable, they have understood what velopharyngeal insufficiency meant; no food regurgitation through the nasal passage was encountered in our sample patients.
Lines 266-306 – the exact repetition of results has been excluded, some explanations have been given for these results and connection to literature data have been mentioned. References have been added throughout the manuscript.

Round 2
Reviewer 3 Report
The manuscript has been improved but I am still not satisfied. My main complaints about this manuscript are as follows:
1. A new co-author has been added. I expect some explanation about the contribution.
2. Enormous number of semicolons used in the text. It results that the sentences are extremely long and difficult to follow the main idea. (Abstract: lines 27-33, Introduction, page 2/line 10, line 28, page 3/line 26, line 36, line 41. Page 4/line 30, line 49.
3. Introduction
· page 2/lines 28-31: preferred by? – references.
· page 2/line 32 – I would replace “neighboring” by “surrounding.”
· page 2/lines 45-46 - I don’t understand.
· Page 2/lines 35-36 are similar to lines 50-51 – this part should be presented in less chaotic way: first the information concerning the current, then the conclusions that bipolar is safer.
· It was mentioned in line 34/page 2 that the group consisted of pediatric patients, line 41 is unnecessary.
· Page 2/lines 28-30 and 42-44 – there is no need to provide the same information twice. Please, chose the proper position in the text.
· Page 2/line 10, line 21 – check reference number.
· Page 3/line 49: informed consent – it is obvious – I assume that You do not perform such procedures without parents’/caregivers’ consent.
· Page 3/line 47 and page 4/lines 3-4: the same information about general anesthesia
· Page 3/line 45: I have mixed feelings about the inclusion criteria. The presented ones are not the same as inclusion criteria for the scientific study but for the surgical procedure.
· Page 4/lines 1nad 6: the same information about 3 surgeons.
· Page 4/line 16: maximum 1 minute – so it could be 5 seconds? Please, be more precise and give some time frame.
4. The Results section is presented in the proper way. It is one of the best parts of this manuscript.
5. Discussion
· Page 9/lines 35-43 – in my opinion this part is unnecessary.
· Page 10/lines 36-43 semi-colons…. It is one sentence… why not to simplify? lines 47-54 – the same as above.
· Page 11 / lines 1-10: the results are presented but they are not referred to other studies.
6. The manuscript is still difficult to read. There is a lot of side plots, the work is not coherent. The discussion did not convincingly emphasize the importance of the conducted research in relation to other publications.
7. In my opinion the manuscript requires deeper changes in the structure and the form of presentation these interesting results. I encourage the authors to reconsider it.
Author Response
1. The new co-author added, Ms. Catalina Voiosu, has worked with the rest of the team from the beginning, being involved in some data collection, analysis. Later on, when she extensively reviewed and edited the article, all the authors have considered she has met the criteria for being a co-author and this is why she has been added. 2. All the semicolons used in the masuscript have been excluded, shorter sentences have been made (abstract: line 27-33, introduction page 2 / line 10, line 28, page 3/ line 26, 36, 41, page 4 / line 30, 49). 3. Introduction - page2 / lines 28-31 - referrences have been added; page 2 / line 32 - the word "neighbouring" has been replaced with surrounding; page 2 / lines 45-46 have been excluded' page 2 / lines 35-36 to 50-51 - the introduction has been reorganised and shortened to sound less chaotic; page 2 / lines 34 and 41 and also page 2 lines 28-30 and 42-44 have been united so that redundant information was eliminated; page 3 / line 49 - we have eliminated redundant information about informed consent; page 3 / line 47 - we have eliminated redundant informations about general anesthesia. Introduction page 3 / line 45 - inclusion criteria have been simplified; in fact inclusion criteria for the scientific study are similar to those for adenoidectomy, we have chosen randomly from adenoidectomy patients until there were 100 patients (we gave questionnaires to patients operated on in 2 days per week during those 3 months). Introduction page 4 / the lines about the 3 selected surgeons were shortened and redundant information has been eliminated; page 4 / line 16 - the information about leaving the package in place for 1 minute has been eliminated and replaced by"until soaked in blood". Results section - has been left unchanged. Discussion - page 9 / lines 35-43 , page 11 /lines 1-10 have been reorganised; every result is discussed referring to literature data, with references. The entire Discussion section has been reorganised, redundant / repetitive information has been excluded. page 10 / lines 36-43, lines 47-54 no more semicolons, shorter sentences. 6. 7. After extensive reviewing the manuscript is easier to read. We discuss the results and compare them to literature data. In fact, we have reconsidered the entire manuscript.
